# Comment on Pata et al. Sclerobanding (Combined Rubber Band Ligation with 3% Polidocanol Foam Sclerotherapy) for the Treatment of Second- and Third-Degree Hemorrhoidal Disease: Feasibility and Short-Term Outcomes. *J. Clin. Med.* 2022, *11*, 218

**DOI:** 10.3390/jcm11092495

**Published:** 2022-04-29

**Authors:** Johannes Jongen, Jessica Schneider, Volker Kahlke, Tilman Laubert

**Affiliations:** Proctological Office Kiel, Department of Proctological Surgery, Park-Klinik, 24105 Kiel, Germany; info@proktologie-kiel.de (J.S.); volker.kahlke@googlemail.com (V.K.); tlaubert@googlemail.com (T.L.)

Dr. Pata kindly tweeted the publication of the above-mentioned paper [1]. We were quite interested because in the abstract, sclerobanding is described as “novel”. In the introduction of the paper, the authors also write the following: “Sclerobanding is a novel technique, recently described in the literature…”. The citation is a video vignette of Colorectal Disease 2021 (by the same authors) [2]. In part 4 of the paper (discussion), the authors write the following: “Five studies [33–37] [3,4,5,6,7], mainly retrospective (Table 3), have been published in the literature about the concomitant use of SCT with RBL in the treatment of HD, …”. The study of the authors is also retrospective, so that is not a novelty. The first cited study was published in 1985 and the last cited in 2003. We wonder why the authors call the technique of simultaneously treating haemorrhoids with rubber bands and sclerosing agent a novelty, although the technique was described/published 35 years ago. To our knowledge, in many proctological offices, combining sclerosing therapy with rubber band ligation has already been practised for a long time, even in our own office since 1982.

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
