# Peer review of "Comment on Pata et al. Sclerobanding (Combined Rubber Band Ligation with 3% Polidocanol Foam Sclerotherapy) for the Treatment of Second- and Third-Degree Hemorrhoidal Disease: Feasibility and Short-Term Outcomes. J. Clin. Med. 2022, 11, 218"

_jcm, 2022, doi:10.3390/jcm11092495_

Round 1
Reviewer 1 Report
The comment is appropriate. The requests for clarification are legitimate considering the previous publications of 1985. In particolar, these two articols reported the same tecnique that Pata et al. describe as new, therefore the comment is fitting.
English language and style not require changes.
Reviewer 2 Report
This is a letter to the editor about the article by Pata F et al. "Sclerobanding (Combined Rubber Band Ligation with 3% Polidocanol Foam Sclerotherapy) for the Treatment of Second- and Third-Degree Hemorrhoidal Disease: Feasibility and Short-Term Outcomes", published in the special issue " Colorectal Surgery: Latest Advances and Prospects".doi: 10.3390/jcm11010218.
All points raised in the letter sound reasonable, but should be published with a mandatory reply by authors of the original article to give a complete overview to the readers.